# Study on Elastic Mixed Mode Fracture Behavior and II-III Coupling Effect

**DOI:** 10.3390/ma16134879

**Published:** 2023-07-07

**Authors:** Xinting Miao, Jinbo Zhang, Haisheng Hong, Jian Peng, Binbin Zhou, Qianqian Li

**Affiliations:** 1School of Mechanical Engineering and Rail Transit, Changzhou University, Changzhou 213164, China; jinbozhang0106@163.com (J.Z.); m18051804661@163.com (H.H.); joepengjian@163.com (J.P.); liqianqian@cczu.edu.cn (Q.L.); 2Jiangsu Key Laboratory of Green Process Equipment, Changzhou University, Changzhou 213164, China; 3Engineering Technology Training Center, Nanjing Vocational University of Industry Technology, Nanjing 210023, China; zhoub_b@163.com

**Keywords:** mixed mode crack, coupling effect, fracture criterion, linear elastic fracture mechanics

## Abstract

Mixed mode fracture is a widely studied topic, while the coupling effects of mixed mode cracking are unclear. In this paper, elastic fracture behaviors and the coupling effects of the mixed mode cracks are studied in detail based on the finite element method, experimental study and linear elastic fracture mechanics. Results show that there always exist II-III coupling effects at the crack tips of mixed mode cracks, which have many effects on the crack tip field and crack propagation behavior. It is found that a mode II component at the tip of a mixed mode crack is the main reason for crack deflection, while the mode III components show no effect. For any mixed mode crack, mode II components at the crack tip can be divided as that by mode II loading which causes plane crack propagation, and by the coupling effect which causes spatial crack propagation. On this basis, a new fracture criterion suitable for any mixed mode crack is proposed, combined with the coupling effect and the linear elastic superposition principle. The research in this paper provides a solution to the problem of an II-III coupling effect in mixed mode fracture research and further promotes the development of fracture mechanics.

## 1. Introduction

In fracture mechanics, there exist three types of fracture modes, including mode I crack (opening mode crack), mode II crack (sliding mode crack) and mode III crack (tearing mode crack), shown in Figure 1. The three basic fracture modes (mode I, mode II and mode III) temporarily or permanently occur in combination, due to either the external loading or the orientation of the crack. Then a non-symmetrical, singular stress field in the vicinity of the crack front is present with the three basic fracture modes in combination. So, the crack grows in a way with an opening, planar or non-planar mode. The stress field in the vicinity of the crack front is defined by the stress intensity factor *K*_I_, *K*_II_ or *K*_III_, shown in Equation (1) [1]. Equations (1a)–(1c) are for plane mixed mode cracks and Equations (1d)–(1f) are for spatial mixed mode cracks. In practice, cases such as cracked structures under complicated loading, kinked or branched cracks, multiple cracks, cracks initiating from notches, cracks in welded or adhesive joints and cracks in composites are with the mixed mode crack problems.
(1a)$$\sigma_{r}=\frac{K_{I}}{4\sqrt{2\pi r}}\left\{5\cos\left(\frac{\varphi }{2}\right)-\cos\left(\frac{3\varphi }{2}\right)\right\}-\frac{K_{\mathrm{II}}}{4\sqrt{2\pi r}}\left\{5\sin\left(\frac{\varphi }{2}\right)-3\sin\left(\frac{3\varphi }{2}\right)\right\}$$
(1b)$$\sigma_{\varphi }=\frac{K_{I}}{4\sqrt{2\pi r}}\left\{3\cos\left(\frac{\varphi }{2}\right)+\cos\left(\frac{3\varphi }{2}\right)\right\}-\frac{K_{\mathrm{II}}}{4\sqrt{2\pi r}}\left\{3\sin\left(\frac{\varphi }{2}\right)+3\sin\left(\frac{3\varphi }{2}\right)\right\}$$
(1c)$$\tau_{r\varphi }=\frac{K_{I}}{4\sqrt{2\pi r}}\left\{\sin\left(\frac{\varphi }{2}\right)+\sin\left(\frac{3\varphi }{2}\right)\right\}+\frac{K_{\mathrm{II}}}{4\sqrt{2\pi r}}\left\{\cos\left(\frac{\varphi }{2}\right)-3\cos\left(\frac{3\varphi }{2}\right)\right\}$$
(1d)$$\tau_{rz}=\frac{K_{\mathrm{III}}}{\sqrt{2\pi r}}\sin\left(\frac{\varphi }{2}\right)$$
(1e)$$\tau_{\varphi z}=\frac{K_{\mathrm{III}}}{\sqrt{2\pi r}}\cos\left(\frac{\varphi }{2}\right)$$
(1f)$$\sigma_{z}=\nu \left(\sigma_{r}+\sigma_{\varphi }\right)=\frac{8\nu }{4\sqrt{2\pi r}}\left\{K_{I}\cos\left(\frac{\varphi }{2}\right)-K_{\mathrm{II}}\sin\left(\frac{\varphi }{2}\right)\right\}$$

All those stress field equations (Equations (1a)–(1f)) are based on a cylindrical coordinate system with the coordinates *r*, *φ* and *z* (shown in Figure 2). For *r* → 0, all stress fields become singular. The parameters *K*_I_, *K*_II_ and *K*_III_ are the stress intensity factors for the three fracture modes.

In order to explain the macroscopic fracture mechanism of the mixed mode cracks from different aspects, researchers have done a lot of research. For I-II mixed mode cracks, studies have shown that there are two different fracture failure mechanisms, the opening type and the shearing type [2]. The process of void nucleation, growth and coalescence usually causes opening fracture failure, and local shear localization usually leads to shear fracture failure [3]. The crack initiation angle follows the maximum circumferential stress criterion (MTS criterion) [4], and the minimum strain energy density criterion (SED criterion) [5] when a crack initiates in an opening mode. While the crack initiation angle tends toward the maximum shear stress direction (MSS criterion) [6], the maximum crack tip opening displacement (COD criterion) shear component direction [7] and the plastic flow direction [8] when a crack initiates in a shear mode. The topic of I-II mixed mode fatigue crack growth has been widely studied, which focused on the crack driving force, the mixed mode fatigue fracture mechanism and the fatigue crack growth prediction model [9,10,11]. 

For I-III mixed mode cracks, a spatial curve surface ahead of the crack tip is present, and there is a torsion angle in the crack propagation path due to the effect of the mode III component. Richard [12], Pook [13] and Schöllmann et al. [14] proposed different fracture criteria to predict the torsion angle, fracture toughness and equivalent stress intensity factor of the crack, including a mode III component. Ayhan [15] studied the I-II-III mixed mode fracture behavior using CTSR specimens. It was considered that the existing criteria underestimate the mixed mode fatigue and fracture life. Yaren [16] et al. studied the I-III mixed mode fatigue crack propagation of 7075-T651 aluminum alloy and found that the MTS criterion [4] can predict the spatial propagation behavior when the mode III components are low, otherwise, it deviates from the test results. Miao [17] studied the mixed mode fracture behaviors of TC4 titanium alloy, and the results showed that the criteria cannot predict the fracture test results well when the mode III component was higher. Richard [18] et al. proposed the corresponding criteria for the case of higher mode III components through PMMA (polymethyl methacrylate). Some of the common criteria mentioned above are listed in Table 1 (which will be studied in the following paper), where the MTS criterion is suitable for plane fracture prediction, and the remaining fracture criteria can be used for spatial three-dimensional fracture prediction (*φ*_0,_
*ψ*_0_ are crack initiation angles, and *K*_IC_ is fracture toughness of mode I crack.). In addition, mixed mode fracture behaviors of composites (like aerospace-grade honeycomb core sandwich composites) have been studied a lot [19,20,21], while the comparisons between fracture behaviors of homogenous elastic material and layered elastic media should be studied further. 

Under mixed mode loading conditions, mode II and mode III fracture modes at the crack tips are coupled with each other [16,22,23], named the “II-III coupling effect”. Specifically, additional mode II components are induced along the crack fronts due to the bending effect of mode III (tearing) loading in the out-of-plane direction; additional mode III components occur by mode II loading, due to the Poisson’s ratio effect of the material. Demir [16] found that the existing fracture criteria deviated from the test results of mixed mode cracking, when the II-III coupling degree was high, and a new criterion was proposed based on the empirical values. The underlying mechanisms acting under general three-dimensional mixed mode loading are complex and highly interactive. Although the coupled modes have been known and observed numerically for many years, they have not been specifically studied. Combined with the criterion above, are the differences between the fracture criterion and test results induced by the coupling effect? What is the effect of the coupling effect on mixed mode fractures and fatigue crack growth, and how much is the effect? All these questions are needed to solve these problems, and the three-dimensional mixed mode fracture problems need to study further. 

On the basis of the above fracture criterion and II-III coupling effect, this paper takes PMMA as the research object, focusing on the study of I-II mixed mode cracks, I-III mixed mode cracks, II-III mixed mode cracks and I-II-III mixed mode cracks. It aims to study the effect of the II-III coupling effect in mixed mode fractures, explore the dominant factors in the propagation behavior of mixed mode cracks, compare the applicability of each fracture criterion and finally propose a unified propagation criterion in order to improve the theory of linear elastic fracture mechanics.

## 2. Research Subjects and Methods

### 2.1. Specimens and Loading Devices

In this paper, the design of the I-II and I-III mixed mode fracture test devices are carried out by referring to the Richard-CTS loading device [24]. The test device mainly includes three parts: the upper and lower chucks connected to the test machine, the arc loading fixture connected to the chucks and the fracture specimen. The specimen studied in this paper is a single-edge notched specimen, the shape and size of the specimen are shown in Figure 3a. The overall loading diagrams are shown in Figure 3b,c. The different mixed mode loading degrees can be realized by the holes at different positions on the arc loading fixture, which are defined by loading angles *β* and *β*′ in Figure 3b,c (0° ≤ *β* ≤ 90°, 0° ≤ *β*′ ≤ 90°). It is a pure mode I loading when *β* or *β′* = 90°, pure mode II loading when *β* = 0°, pure mode III loading when *β*′ = 0° and mixed mode loading when 0° < *β*(*β*′) < 90°. The larger the *β* or *β*′ is, the larger the mode I component at the crack tip is. The mode mixity is defined by the ratio of *K*_I_ and *K*_II_ ahead of the crack tip, that is *K*_II_^n^ = *K*_II_/(*K*_I_ + *K*_II_ + *K*_III_) and *K*_III_^n^ = *K*_III_/(*K*_I_ + *K*_II_ + *K*_III_), which will be discussed in the following paper.

### 2.2. Test Devices

The mixed mode fracture specimen and the mixed mode loading experimental device are shown in Figure 4. The material of PMMA (polymethyl methacrylate) shows a brittle fracture mode [25], which is widely used in linear elastic fracture research, so PMMA is selected as the material in this paper. The material properties are as follows: elastic modulus *E* = 2660 Mpa and Poisson’s ratio *v* = 0.34. The I-II and I-III mixed mode loading devices are shown in Figure 4b,c, respectively. In the device, the specimen is connected to the arc loading fixture with screws, and the upper and lower loading fixtures are connected to the testing device with pins. The force on the test device can be transmitted to the specimen through the connection of each part so that the specimen’s crack tip presents a mixed mode state. High-strength materials are selected for the loading fixture, as well as the pin.

### 2.3. Finite Element Models

The finite element model is consistent with the test device and can also achieve different degrees of mixed mode loading, as shown in Figure 5. The finite element model is also divided into two parts: the loading fixture and the specimen. The contact is set between the loading fixture and the specimen. The red surface in Figure 5 is the main surface, the purple surface is the slave surface and the friction coefficient is set as 0.2. Under the action of contact between each part, the force on the fixture can be transmitted to the specimen so that the mixed mode state at the crack tip can be realized. In order to ensure that the fixture does not deform, the fixture adopts a rigid body. Consistent with the test device, equidistant loading holes are also established on the finite element model of the fixture, and different mixed mode loading degrees can be achieved through these loading holes. In order to consider the singularity of the crack tip, the wedge mesh is used at the crack tip of the fracture specimen, and the minimum mesh size around the crack tip region is 0.1 mm, as shown in Figure 5. The mesh sensitivity of J-integral at the crack tip is shown in Table 2, which shows that the mesh size at the crack tip of 0.1 mm is accurate enough. Also, the above modeling methods have been applied to the study of mixed mode fatigue and the fracture of different materials [2,11,26].

## 3. I-II and I-III Mixed Mode Crack Propagation Behavior

For a three-dimensional structure with a crack, the mechanical behavior at the crack tip is an important factor affecting the propagation behavior. In this paper, the stress intensity of factor *K*, the crack tip field and the propagation behavior of mixed mode cracks are systematically studied.

### 3.1. The Distribution of K Factor along the Thickness Direction

Figure 6 shows the distribution of stress intensity on factor *K* along the thickness direction at the crack tip of the CTS specimen and the CTT specimen under I-II and I-III mixed mode loading. It can be seen that there are mode I, mode II and mode III components at the crack tips simultaneously. This phenomenon has been proposed in previous studies defined as the “II-III coupling effect” [16,17]. That is, mode II or mode III loading will produce the coupled mode III or mode II component. For the CTS specimen under I-II mixed mode loading, the maximum mode III component produced by the coupling effect at the crack tip is about half of the mode II component, as shown in Figure 6a. Similarly, for the CTT specimen with I-III mixed loading applied, in addition to mode I and mode III components caused by the loading, there also exists a mode II component generated by the coupling effect. The mode II component is anti-symmetric about the middle surface of the specimen, with the maximum mode II component generated by the crack tip coupling approximate to the mode III component, as shown in Figure 4b. Therefore, the CTS specimen and CTT specimen under the I-II and I-III mixed loadings cannot be simply regarded as pure I-II and I-III mixed cracks, the II-III coupling effect at the crack tip needs to be considered. In order to facilitate the subsequent description, the stress intensity factors generated by mode II and mode III loading are defined as *K*_II_^A^ and *K*_III_^A^, and the stress intensity factors generated by the II-III coupling effects are defined as *K*_III_^C^ and *K*_II_^C^.

### 3.2. Stress Field Analyses

In order to figure out the effect of the coupling effect, stress distributions for specimens under mode II and III loadings are present based on the finite element method, including *σ*_rr_, *σ*_θθ_, *σ*_zz_, *τ*_rθ_, *τ*_rz_ and *τ*_θz_, which are shown in Figure 5a,b, respectively. Also, the expressions of each stress component are shown in Equations (1a)–(1f). Firstly, for the CTS specimen under pure mode II loading, the difference in the crack tip field between the middle surface and the outer surface is mainly due to *τ*_rz_ and *τ*_θz_. Combined with Equations (1a)–(1f), the difference of the cracked tip field along the thickness is due to the mode III component coupled by the mode II loading. The existence of the mode III component will generate additional shear stress (*τ*_rz_ and *τ*_θz_). The specific stress distribution is shown in Figure 7a. For the CTT specimen under pure mode III loading, the difference in the crack tip field between the center surface and the surface is mainly due to *σ*_rr_, *σ*_θθ_, *σ*_zz_ and *τ*_rθ_. Similarly, combined with the expression of Equations (1a)–(1f), the difference in the crack tip field along thickness is due to the mode II component coupled by mode III loading. The existence of the mode II component will generate additional shear stress components. The specific stress distribution is shown in Figure 7b.

### 3.3. Crack Growth Behavior

(1)Analyses of experimental results

Mixed mode fracture tests were carried out on the CTS and CTT specimens, and the crack growth paths and load-displacement curves were obtained. For the I-II mixed mode fracture, it presents a plane propagation trajectory, and the higher the mode II component, the larger the deflection angle and the higher the *L*-*D* curve. This is mainly because the greater the crack deflection angle (*φ*_0_) is, the greater the energy required, that is, the higher the ultimate bearing capacity. For the I-III mixed mode fracture, it presents a spatially anti-symmetric propagation trajectory. The higher the mode III component, the greater the spatial torsion degree (*ψ*_0_) is, and the higher the ultimate bearing capacity. The specific test results are shown in Figure 8.

Also, the *L*-*D* curves from the FEM (finite element method) are obtained in Figure 8 to compare with the test results. The rising slopes of the *L*-*D* curves are all that, (*β* or *β*′ = 15°) < (*β* or *β*′ = 45°) < (*β* or *β*′ = 90°), which validates that the FEM results are accurate indirectly. It should be noted that: (i) it does not allow crack growth by the FEM model, which is why the FEM results are higher than that by the test; (ii) due to the space between the loading pin and loading clamp, the *L*-*D* curves by test show the non-linear changing trend firstly.

(2)Comparison between experimental study and theoretical solution

According to the above research, it can be seen that mode II loading causes plane crack growth (crack growth angle *φ*_0_), as shown in Figure 6. The mode III component causes spatial crack growth, and it presents a continuous spatial expansion trajectory, as shown in Figure 8 and Figure 9a. The spatial curved propagation trajectory can also be regarded as that synthesized by the propagation trajectories on different planes along the crack front direction, as shown in Figure 9b. On this basis, in order to facilitate the study of fracture mechanics, the spatial surface expansion trajectory is simplified. The final simplified diagram is shown in Figure 9c, and the spatial propagation angle is defined as *ψ*_0_.

Based on the evidently mixed mode, brittle-fracture characteristics in Figure 8, on the basis of the above content, the experimental solution is compared with the theoretical solution (fracture criterion in Table 1), as shown in Figure 10. For the I-II mixed mode crack, the growth angle (*φ*_0_) and fracture toughness (*K*_I_/*K*_IC_ − *K*_II_/*K*_IC_) are in good agreement with the fracture criteria, and the differences between the criteria are also small. For the I-III mixed mode crack, Sih’s criterion has some differences from the other criterion. It is considered that there is no deflection with the mode III component in Sih’s criterion. From the comparison in Figure 7c,d, the test results are consistent with Pook’s criterion. It should be noted that the mixed mode fracture toughness of PMMA in this paper was determined using a reference to the method of determining the plane strain fracture toughness (*K*_IC_) in the standard [27], which was also applied to the mixed mode fracture research [17].

A discussion on spatial crack propagation behavior is shown in Figure 10, the criteria can more or less predict the I-II and I-III mixed mode fracture accurately, except for Sih’s criterion. But it should be noted that: (i) The spatial propagation angle of *ψ*_0_ in Figure 10 is a simplified angle in Figure 9c, while the actual propagation trajectory of the mode III crack is a continuous spatial curved surface. (ii) According to the distribution of the *K* factor in Figure 6, the mode II component and mode III component are coupled with each other. Especially for the CTT specimen under I-III mixed mode loading, the II-III coupling degree is high, and the coupled mode II component cannot be ignored. So, what is the effect of the coupled mode II component? (iii) The energy-based Sih’s criterion holds that the mode III component does not cause the deflection of the crack, and only the mode II component causes the deflection of the crack. Is the spatial propagation angle of the CTT specimen under I-III mixed mode loading caused by the coupling mode II component?

From the propagation behaviors of the I-II mixed mode crack and the I-III mixed mode crack in Figure 8, there are plane propagation angles (*φ*_0_) of the I-II mixed mode crack, and spatial propagation angles (*ψ*_0_) of the I-II mixed mode crack. Also, the propagation trajectory of the crack is basically consistent with the distribution of the *K*_II_ component at the crack tip, as shown in Figure 11a. It is considered that the mode II component is the main reason for the crack deflection. For the CTS specimen under I-II mixed mode loading, *K*_II_^A^ caused by loading is the main reason for the plane crack propagation. While for the CTT specimen under I-III mixed mode loading, *K*_II_^C^ generated by the coupling effect is the main reason for the spatial crack propagation, as shown in Figure 11a.

In order to further quantify the degree of the II-III coupling effect, the coupling factor *C* is defined as *C* = *K*_II_/*K*_III_. For mode II loading, *K*_II_ is the average value of *K* along the thickness direction, expressed as *K*_II_^ave^, and *K*_III_ is that at the outer surface of the specimen, expressed as *K*_III_^out^, that is, *C*^I-II^ = *K*_II_^ave^/*K*_III_^out^. For mode III loading, *K*_III_ is the average value of *K*_III_^ave^ along the thickness direction, and *K*_II_ is that at the outer surface of the specimen, expressed as *K*_II_^out^, that is, *C*^I-III^ = *K*_II_^out^/*K*_III_^ave^. Through calculation and studying, it is found that for certain specimen and loading styles, *C* remains unchanged; for example, for any CTS specimen under I-II mixed loading, *C* ≈ 2.8, for any CTT specimen under I-III combined loading, *C* ≈ 1.

In order to overcome the error and inconvenience caused by the simplification of the spatial propagation angle in Figure 9c, this paper believes that it is more suitable using the outermost plane growth angle (*φ*_0_^out^ shown in Figure 9b) to describe the spatial propagation angle (*ψ*_0_), which can better reflect the expansion process of the spatial continuous curved surface, as shown in Figure 9b. The crack propagation angles (*φ*_0_^out^) at the outermost layers of the specimens by tests are shown in Figure 11b; compared with the MTS criterion, where *φ*_0_ is equal to *φ*_0_^out^ for the CTS specimen, and *ψ*_0_ is instead by *φ*_0_^out^ for the CTT specimen. Results show that the crack propagation angles are in good agreement with the MTS criterion and related to *K*_II_/(*K*_I_ + *K*_II_) or *K*_III_/(*K*_I_ + *K*_III_).

## 4. Study on II-III and I-II-III Mixed Mode Crack Propagation Behavior

### 4.1. Fracture Specimen

The specimens and loading fixtures studied in this section are shown in Figure 12. Figure 12a shows the CST specimen (compact shearing tearing specimen) under II-III mixed mode loading, the loading mode and contact setting and meshing are consistent with the I-II and I-III mixed mode loading devices above. Figure 12b shows the cracked specimen under I-II-III mixed mode loading, the loading is applied to the holes at both ends of the specimen. The displacement loads along the x, y and z directions are defined as *U*x, *U*y and *U*z, respectively. In order to achieve different degrees of I-II-III mixed mode loading, this paper takes the following four loading methods: *U*x/*U*y = 0.5, *U*y/*U*z = 0.5; *U*x/*U*y = 0.5, *U*y/*U*z = 1; *U*x/*U*y = *U*y/*U*z = 0.5; and *U*x/*U*y = 1, *U*y/*U*z = 1. It should be noted that all the fracture specimens under different mixed mode loadings are consistent.

### 4.2. Analyses of Mixed Mode Crack Propagation Behavior

#### 4.2.1. Study on the Distribution of K Factor

Figure 13a shows the *K* distribution at the crack tip of the CST specimen under II-III mixed loading. With the increase of the loading angle (*β*″), *K*_I_ keeps 0, *K*_II_ gradually changes from symmetrical distribution to anti-symmetric distribution about the middle plane, and *K*_III_ gradually changes from anti-symmetric distribution about the central plane to symmetrical distribution. On the other hand, it gradually changes from pure mode II to pure mode III loading applied on the specimen. Figure 13b shows the *K* distribution at the crack tip of the specimen under I-II-III combined loading. *K*_I_ is symmetrical about the central plane, and both *K*_II_ and *K*_III_ are asymmetrical about the central plane. The distribution law is related to the loading components applied to the specimen.

It has been concluded that the mode II or mode III component at the crack tip will produce the corresponding coupled mode III or mode II component. Then, for *K*_II_ and *K*_III_ in Figure 13, how do we distinguish *K*_II_^A^ and *K*_III_^A^ caused by the applied loading, as well as *K*_II_^C^ and *K*_III_^C^ caused by the coupling effect? For II-III mixed mode crack and I-II-III mixed mode crack, what is the influence of the coupling effect?

#### 4.2.2. Theoretical Analyses and Discussions

(1)Theoretical analyses of II-III mixed mode crack

According to the above analyses and the linear elastic superposition principle in linear elastic fracture mechanics, the *K* factor at the crack tip is induced by both the loading and coupling effect for any II-III mixed mode crack, and the specific decomposition process of the *K* factor is shown in Figure 14. *K* factor meets the following relations: *K*_II_ = *K*_II_^A^ + *K*_II_^C^ and *K*_III_ = *K*_III_^A^ + *K*_III_^C^. Moreover, as described in Figure 11a, the crack propagation angle of the II-III mixed mode crack can be divided into the plane crack deflection angle (*φ*_0_) caused by *K*_II_^A^ and the spatial crack deflection angle (*ψ*_0_) caused by *K*_II_^C^.

Based on the above linear elastic superposition principle, the propagation angle of the II-III mixed mode crack can be obtained only by obtaining the proportional relationship between the mode II and mode III components, which is expressed as Equation (2a). Moreover, the spatial propagation angle (*ψ*_0_ ^III^) is represented by the outer surface propagation angle (*φ*_0_^out^) (Figure 9b), and the II-III mixed mode crack propagation angle can be expressed as Equation (2b).
(2a)$$\left(\varphi_{0}^{II-III},\;\psi_{0}^{II-III}\right)\;=\;\left(A\;\times \;\varphi_{0}^{II},\;B\;\times \;\psi_{0}^{\text{III}}\right)$$
(2b)$$\left(\varphi_{0}^{II-III},\;\psi_{0}^{II-III}\right)\;\to \;\left(\varphi_{0}{}^{II-III},\varphi_{0}{}^{II-III-out}\right)\;=\;\left(A\;\times \;\varphi_{0}^{II},\;B\;\times \;\varphi_{0}^{\text{III-out }}\right)$$
where *φ*_0_^II^ and *ψ*_0_^III^ (*φ*_0_^III-out^) are the crack propagation angles (propagation angle at outermost layer) of the pure mode II and pure mode III, respectively, and *φ*_0_^II^ and *φ*_0_^III-out^ both can be obtained by MTS criterion in Figure 11b. *A* and *B* are functions related to *K*_II_/(*K*_II_ + *K*_III_) and *K*_III_/(*K*_II_ + *K*_III_), respectively, which will be obtained in the following.

(2)Theoretical analyses of I-II-III mixed mode crack

For the I-II-III mixed mode crack, it can be seen as a linear superposition of mode I, mode II and mode III crack. The distribution of the *K* factor at the crack tip can be decomposed as shown in Figure 13. The *K* factor in the figure also satisfies the following relationships: *K*_II_ = *K*_II_^A^ + *K*_II_^C^ and *K*_III_ = *K*_III_^A^ + *K*_III_^C^. The spatial propagation angle of the I-II-III mixed mode crack can also be divided into the plane crack deflection angle (*φ*_0_) caused by *K*_II_^A^ and the spatial deflection angle (*ψ*_0_) caused by *K*_II_^C^.

Based on the linear superposition relationship of mode I, mode II and mode III, it is also only necessary to obtain the proportional relationship between mode II and mode III (*K*_II_/(*K*_I_ + *K*_II_
*+ K*_III_) and *K*_III_/(*K*_I_ + *K*_II_
*+ K*_III_)), then the crack propagation angle can be obtained. The expression of the crack propagation angle is consistent with Equations (2a) and (2b) for the II-III mixed mode crack. It should be noted that *A* and *B* are the functions related to *K*_II_/(*K*_I_ + *K*_II_ + *K*_III_) and *K*_III_/(*K*_I_ + *K*_II_ + *K*_III_), respectively, which will be obtained in the following.

The *K* factor mentioned above also satisfies the following relationship (Figure 14 and Figure 15):

(i)Since *K*_II_^C^ and *K*_III_^C^ are anti-symmetric about the center points, *K*_II_^A^ ≈ *K*_II_^ave^ ≈ $\frac{1}{B}\int_{0}^{B}K_{II}dz$, *K*_III_^A^ ≈ *K*_III_^ave^ ≈ $\frac{1}{B}\int_{0}^{B}K_{III}dz$;

(ii)The coupling factor (*C*^II^ and *C*^III^) is calculated as follows: *C*^II^ = *K*_II_^A^/*K*_III_*^C^*^-out^ ≈ *K*_II_^A^/(*K*_III_^out^ − *K*_III_^A^) and *C*^III^ = *K*_II_*^C^*^-out^/*K*_III_^A^ ≈ (*K*_II_^out^ − *K*_II_^A^)/*K*_III_^A^, where *K*_i_^A^ and *K*_i_^out^ are the *K* factors induced by loading and at the outer surface, which both can be obtained by finite element method (i = II, III);

(iii)Note that the *A* and *B* above are functions related to *K*_II_/(*K*_I_ + *K*_II_ + *K*_III_) and *K*_III_/(*K*_I_ + *K*_II_ + *K*_III_), respectively, where *K*_I_, *K*_II_ and *K*_III_ are caused by loading, denoted by *K*_I_^A^, *K*_II_^A^ and *K*_III_^A^, or the average value of *K* along the thickness direction (*K*_I_^ave^, *K*_II_^ave^, *K*_III_^ave^). For convenience, it is considered that *K*_i_ = *K*_i_^A^ = *K*_i_^ave^ (i = I, II, III).

### 4.3. Discussion and Verification of the New Criterion

In this section, the criterion applicable to any I-II-III mixed mode crack is proposed based on the above conclusions. The specific analyzing process is shown in Figure 16. For the pure mode II or mode III crack, the plane crack angle (*φ*_0_^II^) and the spatial angle (*φ*_0_^III^) (*φ*_0_^II-out^) can be obtained by the MTS criterion (shown in Figure 11b). For any mixed mode crack, the crack deflection angle is related to the proportion of each crack tip component, that is *K*_II_^n^ = *K*_II_/*K*_I_ + *K*_II_ + *K*_III_ and *K*_III_^n^ = *K*_III_/*K*_I_ + *K*_II_ + *K*_III_, where the plane crack angle can be obtained by the value of *A* × *φ*_0_^II^ and the spatial angle can be obtained by the value of *B* × *φ*_0_^III^ (shown in Figure 14 and Figure 15). Therefore, any mixed mode crack propagation angle can be expressed as:


(3a)
$$\left(\varphi_{0}^{I-II-III},\;\psi_{0}^{I-II-III}\right)\;=\;\left(A\;\times \;\varphi_{0}^{II},\;B\;\times \;\psi_{0}^{\text{III}}\right)\to \left(A\;\times \;\varphi_{0}^{II},\;B\;\times \;\varphi_{0}^{\text{III-out }}\right)$$



(3b)
$$A\;=\;2\;\times \;\left(K_{{\mathrm{II}}^{n}}\right)\;-\;{\left(K_{{II}^{n}}\right)}^{2}$$



(3c)
$$B\;=\;2\;\times \;\left(K_{{\mathrm{III}}^{n}}\right)\;-\;{\left(K_{{III}^{n}}\right)}^{2}$$


The functions of *A* and *B* with respect to *K*_II_^n^ and *K*_III_^n^ are obtained by experimental solutions of the I-II mixed mode crack and the I-III mixed mode crack, respectively, in Figure 11b. The propagation criteria of different mixed mode cracks, based on the above analyses, are summarized in Table 3.

In order to prove the accuracy of the above criteria (Equations (3a)–(3c)), the new criteria proposed in this paper are compared with the existing criteria (MPS criterion, Richard’s criterion). The final propagation angles (*φ*_0_ and *φ*_0_^out^) are shown in Figure 17a,b, respectively. It should be noted that the spatial angle in the existing criterion is *ψ*_0_, while the spatial angle (*ψ*_0_) is converted to the plane angle (*φ*_0_^out^) at the outer surface by the geometric relationship of the specimen’s size in Figure 1. The conversion process can be referred to in Figure 11b, where the dimension of *L* is set as 2.5 mm, which is smaller compared with the ligament and consistent with the test results. It can be found from the figure that the criterion proposed in this paper is consistent with Richard’s criterion, the MPS criterion and the experimental values. Therefore, it can be proved that the new criterion proposed in this paper has a certain accuracy in evaluating the linear elastic fracture behavior.

## 5. Conclusions

(1)The cracked specimen will produce the coupling effect at the crack tip, with I-II mixed mode or I-III mixed mode loading applied. The coupling component is antisymmetric with respect to the middle plane of the specimen. For the I-II mixed mode crack, the coupling degree is low, while for the I-III mixed mode crack, the coupling degree is high. Moreover, the mixed mode crack tip field is greatly affected by the coupling effect, there are great differences in the crack tip field between the middle surface and the outer surface of the specimen.(2)The I-II mixed mode fracture criterion can accurately predict the fracture behavior of the CTS specimens under I-II mixed mode loading, and the coupling effect has little effect. However, there are great differences between the I-III mixed mode fracture criteria. The Richard criterion is in good agreement with the experimental results, while the coupling effect cannot be ignored.(3)The mode II component is the main cause of crack deflection. The *K*_II_^A^ caused by loading causes the plane expansion (*φ*_0_), and the *K*_II_^C^ caused by coupling causes the spatial expansion (*ψ*_0_). The spatial propagation angle (*ψ*_0_) can be described by the propagation angle (*φ*_0_^out^) at the outer surface.(4)The II-III mixed mode loading and I-II-III mixed mode loading also have the II-III coupling effect, and also meet the rule of “*K*_II_^A^ by loading causes the plane crack propagation and *K*_II_^C^ by the coupling effect causes spatial crack propagation”. Based on the above conclusions, a propagation criterion suitable for any I-II-III mixed mode crack is proposed, which can accurately evaluate the propagation angle of the mixed mode crack. Moreover, in addition to the crack initiation angle, fracture toughness is also an important factor at the crack tip, the prediction of which, considering the coupling effect, will be discussed in our next study.

## Figures and Tables

**Figure 1 materials-16-04879-f001:**
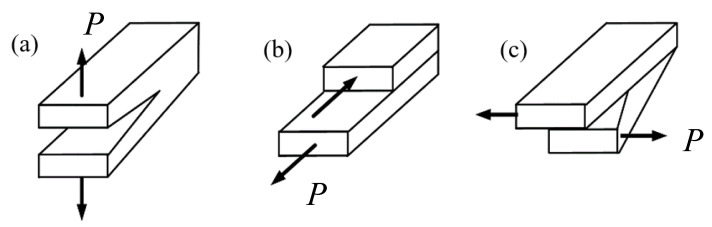
Fracture mode: (**a**) mode I crack, (**b**) mode II crack and (**c**) mode III crack.

**Figure 2 materials-16-04879-f002:**
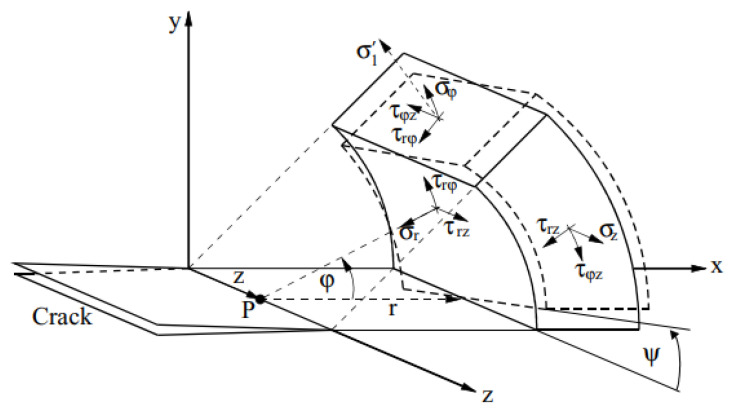
Schematic diagram of linear elastic composite crack tip stress field [1].

**Figure 3 materials-16-04879-f003:**
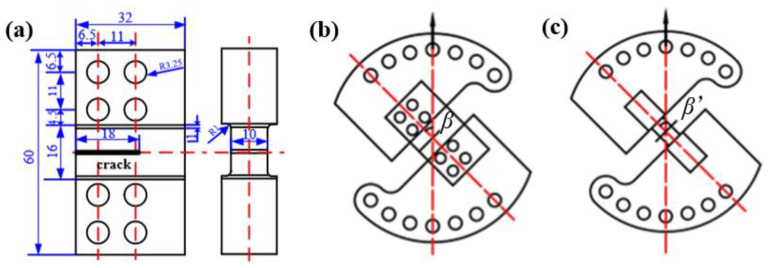
Diagram of the mixed mode loading device. (**a**) Fracture specimen (unit: mm), (**b**) I-II mixed mode loading and (**c**) I-III mixed mode loading.

**Figure 4 materials-16-04879-f004:**
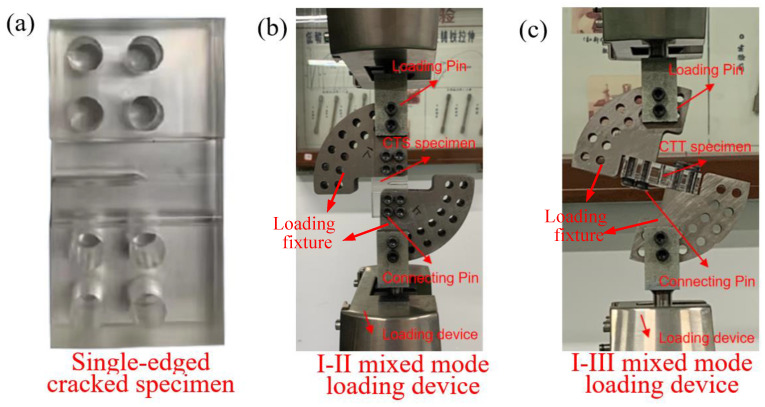
Physical view of the test set-up. (**a**) Fracture specimen, (**b**) I-II mixed mode loading and (**c**) I-III mixed mode loading.

**Figure 5 materials-16-04879-f005:**
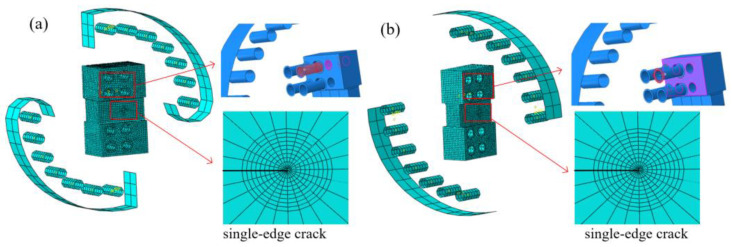
Finite element models: (**a**) specimen under I-II mixed mode loading, (**b**) specimen under I-III mixed mode loading.

**Figure 6 materials-16-04879-f006:**
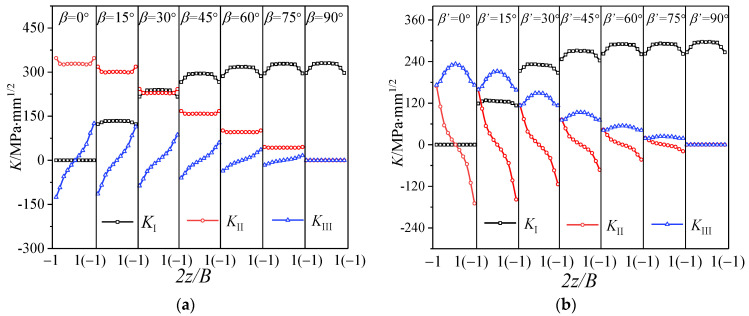
Distribution of *K* factor: (**a**) I-II mixed mode crack (CTS) and (**b**) I-III mixed mode crack (CTT).

**Figure 7 materials-16-04879-f007:**
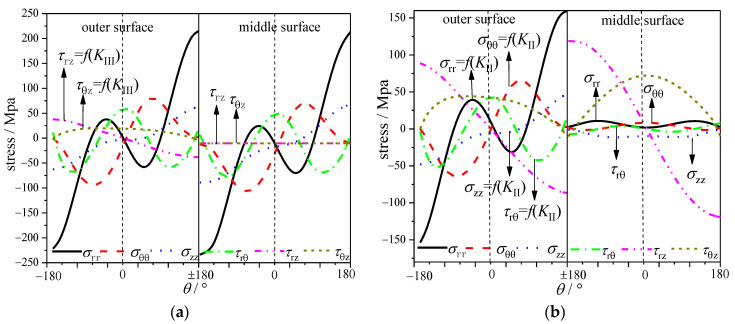
Stress distributions: (**a**) CTS specimen under mode II loading and (**b**) CTT specimen under mode III loading.

**Figure 8 materials-16-04879-f008:**
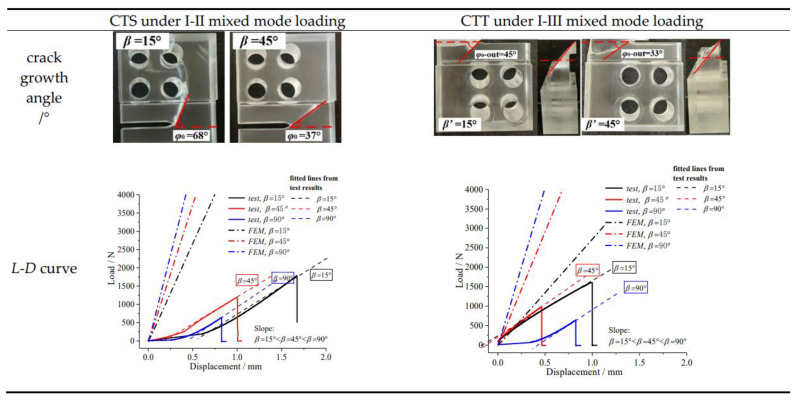
Mixed mode crack propagation test results.

**Figure 9 materials-16-04879-f009:**
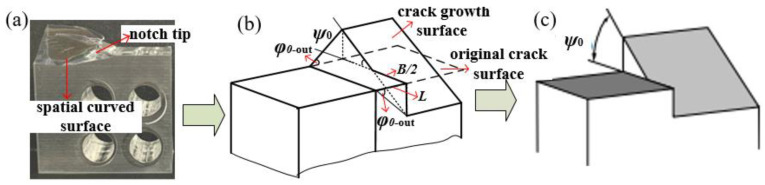
The simplified process diagram of space propagation angle. (**a**) fracture trajectory; (**b**) Sketch Map of fracture trajectory; (**c**) the simplified fracture trajectory.

**Figure 10 materials-16-04879-f010:**
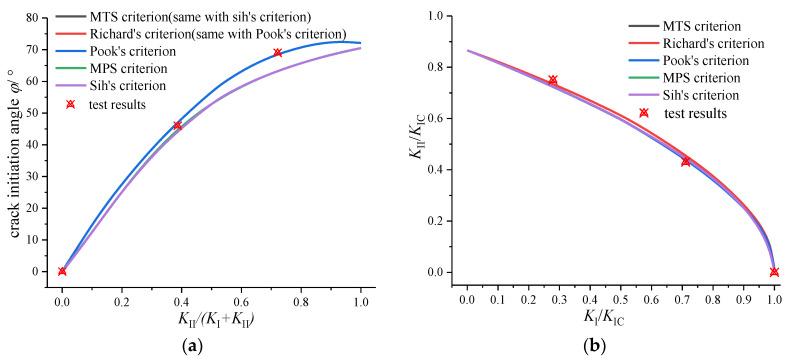
Comparison between test results and fracture criterion: (**a**) φ of I-II mixed mode crack, (**b**) *K*_IC_-*K*_IIC_ of I-II mixed mode crack, (**c**) *ψ* of I-III mixed mode crack and (**d**) *K*_IC_-*K*_IIIC_ of I-III mixed mode crack.

**Figure 11 materials-16-04879-f011:**
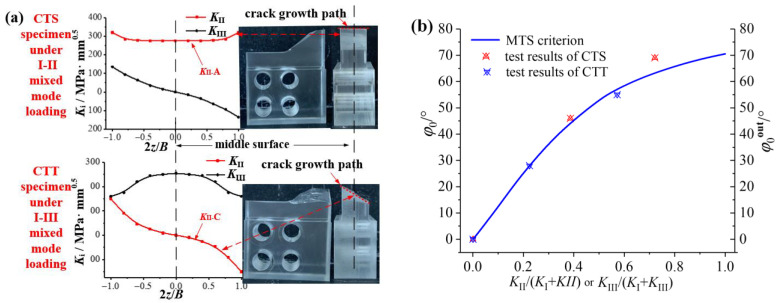
Spatial propagation angle of mixed mode crack: (**a**) comparison between *K* distribution and crack growth path and (**b**) crack propagation angle at the outmost layer *φ*_0_^out^.

**Figure 12 materials-16-04879-f012:**
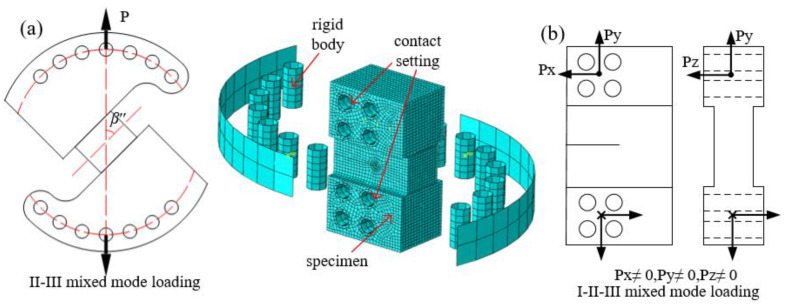
Study sample diagram and finite element model: (**a**) CST specimen under II-III mixed mode loading and (**b**) specimen under I-II-III mixed mode loading.

**Figure 13 materials-16-04879-f013:**
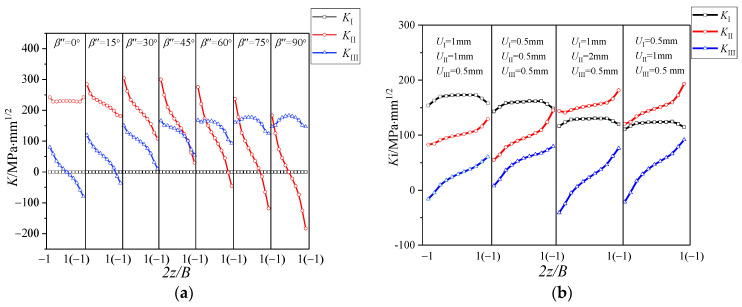
Distribution of *K* factor: (**a**) II-III mixed mode loading and (**b**) I-II-III mixed mode loading.

**Figure 14 materials-16-04879-f014:**
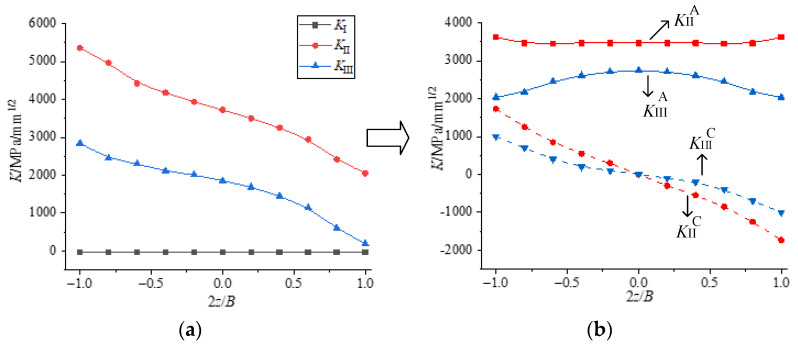
The decomposition process of *K* factor for II-III mixed mode crack. (**a**) distribution of *K*; (**b**) decomposition of *K*.

**Figure 15 materials-16-04879-f015:**
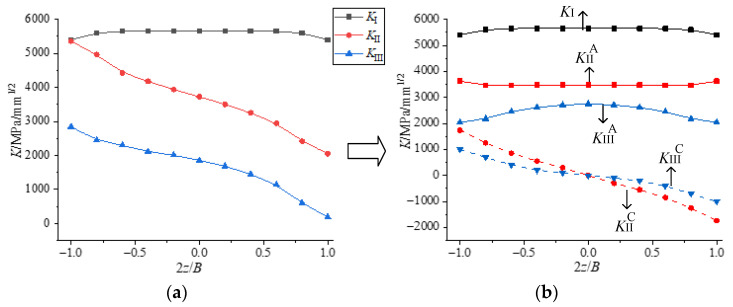
Decomposition process of *K* factor for I-II-III mixed mode crack. (**a**) distribution of *K* (**b**) decomposition of *K*.

**Figure 16 materials-16-04879-f016:**
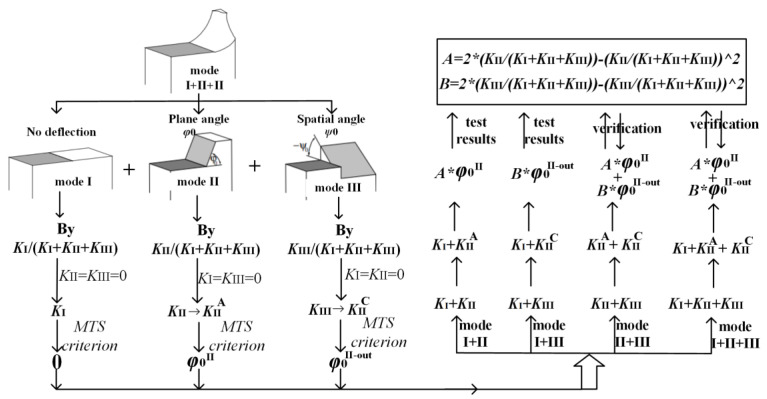
Analyzing process diagram of mixed mode crack propagation criterion.

**Figure 17 materials-16-04879-f017:**
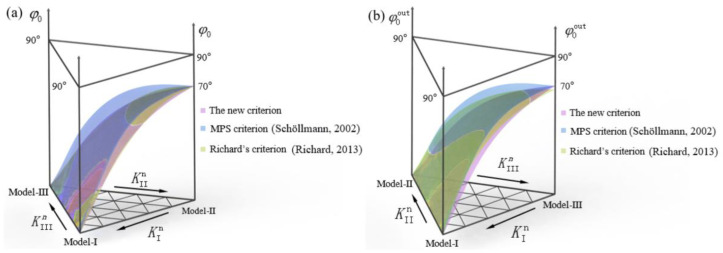
Comparison of the new criterion considering coupling effect with the existing criterion. (**a**) Crack initiation angle *φ*_0_ of I-II mixed mode loading, (**b**) Crack initiation angle *φ*_0_^out^ of I-III mixed mode loading [13,18].

**Table 1 materials-16-04879-t001:** Linear elastic fracture criterion.

Presenter	Description	*φ*_0_/*ψ*_0_/*K*_IC_ Models/Expression
Erdogan and Sih [4] (MTS criterion)	Maximum tangential stress criterion	$\phi_{0}=-\arccos\left(\frac{3K_{II}^{2}+K_{I}^{}\sqrt{K_{I}^{2}+8K_{II}^{2}}}{K_{I}^{2}+9K_{II}^{2}}\right)$ $K_{IC}=\cos\frac{\phi_{0}}{2}\left[K_{I}\cos^{2}\frac{\phi_{0}}{2}-\frac{3}{2}K_{II}\sin\phi_{0}\right]$
Schollmann [13] (MPS criterion)	Maximum principal stress criterion	$\sigma \prime_{1}=\frac{\sigma_{\varphi }+\sigma_{z}}{2}+\frac{1}{2}\sqrt{{\left(\sigma_{\varphi }-\sigma_{z}\right)}^{2}+4\tau^{2}{}_{\varphi z}}$ ${\left.\frac{\partial \sigma_{1}^{\sigma }}{\partial \varphi }\right|}_{\varphi =\varphi_{0}}=0,{\left.\frac{\partial^{2}\sigma_{1}^{\sigma }}{\partial \varphi^{2}}\right|}_{\varphi =\varphi_{0}}=0$
Sih [5]	Strain energy density criterion	$S=a_{11}K_{I}{}^{2}+2a_{12}K_{I}K_{II}+a_{22}K_{II}{}^{2}+a_{33}K_{III}{}^{2}$ ${\left.\frac{\partial S}{\partial \varphi }\right|}_{\varphi =\varphi_{0}}=0,{\left.\frac{\partial S}{\partial \psi }\right|}_{\psi =\psi_{0}}=0$
Richard [18]	an equivalent stress intensity factor *K*_eq_ is defined comparable to the equivalent stress *σ*_eq_	$K_{IC}=\frac{K_{I}}{2}+\frac{1}{2}\sqrt{K_{I}^{2}+4{\left(\alpha_{1}K_{II}\right)}^{2}+4{\left(\alpha_{2}K_{III}\right)}^{2}}$ $\phi_{0}=\mp \left[155.5{}^{\circ}\frac{\left|K_{II}\right|}{\left|K_{I}\right|+\left|K_{II}\right|}\right]-{\left[83.4{}^{\circ}\frac{\left|K_{II}\right|}{\left|K_{I}\right|+\left|K_{II}\right|}\right]}^{2}$ $\psi_{0}=\mp \left[C\frac{\left|K_{II}\right|}{K_{I}+\left|K_{II}\right|+\left|K_{III}\right|}+D{\left(\frac{\left|K_{II}\right|}{K_{I}+\left|K_{II}\right|+\left|K_{III}\right|}\right)}^{2}\right]$
Pook [10]		$K_{eqI,II}=\frac{0.83K_{I}+\sqrt{0.4489K_{I}^{2}+3K_{II}^{2}}}{1.5}$ $K_{IC}=\frac{K_{\mathrm{eq}I,II}\left(1+2\nu \right)+\sqrt{K_{\mathrm{eq}I,II}^{2}{\left(1-2\nu \right)}^{2}+4K_{III}}}{2}$

**Table 2 materials-16-04879-t002:** *J*-integral of CTS specimen under I-II mixed mode loading (*β* = 60°).

Minimum Mesh Size at Crack Tip/mm	0.1	0.2	0.3	Error %
*J*-integral at outer surface/N/mm	3.634	3.609	3.621	0.6
*J*-integral at middle surface/N/mm	4.142	4.15	4.154	0.2

**Table 3 materials-16-04879-t003:** Summary of mixed mode crack propagation criterion.

Crack Type	Coupling Factor *C*	Plane Propagation Angle *φ*_0_/°	Spatial Propagation Angle *ψ*_0_ *→ φ*^out^/°
mode I	*C*^I^ = *K*_II_/*K*_III_ = 1	0	0
mode II	*C*^II^ = *K*_II_/*K*_III_ > 1	*A* × *φ*_0_^II^, *A* = 1	0
mode III	*C*^III^ = *K*_II_/*K*_III_ ≤ 1	0	*ψ*_0_ → *φ*_0_^out^, *B* × *φ*_0_^III-out^, *B* = 1
I-II	*C*^I-II^ = *C*^II^ =*K*_II_/*K*_III_ > 1	*A* × *φ*_0_^II^	0
I-III	*C*^I-III^ = *C*^III^ =*K*_II_/*K*_III_ ≤ 1	0	*ψ*_0_ → *φ*_0_^out^, *B* × *φ*_0_^III-out^
II-III	*C*^II^ = 1 *C*^III^ = *K*_II_/*K*_III_	*A* × *φ*_0_^II^	*ψ*_0_ → *φ*_0_^out^, *B* × *φ*_0_^III-out^
I-II-III	*C*^II^ = 1 *C*^III^ = *K*_II_/*K*_III_	*A* × *φ*_0_^II^	*ψ*_0_ → *φ*_0_^out^, *B* × *φ*_0_^III-out^

## Data Availability

Not applicable.

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
