# Peer review of "Study on Elastic Mixed Mode Fracture Behavior and II-III Coupling Effect"

_materials, 2023, doi:10.3390/ma16134879_

Round 1
Reviewer 1 Report
This article investigated the material's mixed-mode fracture behavior and studied the coupling effect. The article provides valuable information to the fracture mechanics research field. However, some minor issues need to be addressed before it can be accepted for publication.
1. Title: it is unclear what type of material is used to evaluate the mixed mode of fracture behavior. Please specify the material.
2. Abstract: please highlight the current issue in this research topic and the purpose of the study. Moreover, the description of the work investigated in this study is shallow and it should be explained in detail.
3. Introduction: I suggest adding a graphical representation of three basic fracture modes and differences.
4. Are the Equations 1a to 1g proposed by the authors?. If not, please provide citations. Furthermore, the notations used in these equations are defined properly.
5. Section 2.1: what is the β value corresponding to mode III. How does the mixed mode β angles are determined?
6. Figure 1: why the curved specimens were used instead of a flat specimens to assess fracture behavior?. Details of the curvature are not presented in the figure.
7. Figure 14 could be explained in more detail.
8. The discussion and conclusion sections are clearly presented. However, the scope for future work may include at the end of the introduction.
Author Response
Dear editor:
Thank you for giving another chance to improve the paper. The responses to reviewers are listed as below.
Reviewer #1
This article investigated the material's mixed-mode fracture behavior and studied the coupling effect. The article provides valuable information to the fracture mechanics research field. However, some minor issues need to be addressed before it can be accepted for publication.
1.Title: it is unclear what type of material is used to evaluate the mixed mode of fracture behavior. Please specify the material.
Response: The material used in this paper is PMMA (Polymethyl methacrylate), which shows a brittle fracture mode (shown in Section 2.2). So, the title has been revised as: “Study on elastic mixed mode fracture behavior and II-III coupling effect”.
2.Abstract: please highlight the current issue in this research topic and the purpose of the study. Moreover, the description of the work investigated in this study is shallow and it should be explained in detail.
Response: According to the reviewer’s suggestion, the abstract has been improved.
3.Introduction: I suggest adding a graphical representation of three basic fracture modes and differences.
Response: A graphical representation of three basic fracture modes has been added, just shown as Fig. 1 in the revised paper.
4.Are the Equations 1a to 1g proposed by the authors?. If not, please provide citations. Furthermore, the notations used in these equations are defined properly.
Response: The citation of Eq 1 has been added, and the notations in the equations have been defined. In addition, a schematic diagram of crack tip stress field (Fig. 2) has been added, in order to better understand the equation.
5.Section 2.1: what is the β value corresponding to mode III. How does the mixed mode β angles are determined?
Response: There were some errors in the original paper, the descriptions have been corrected and improved, just shown in the revised paper. “It is pure mode I loading when β or β’=90°, pure mode II loading when β=0°, pure mode III loading when β’=0°, and mixed mode loading when 0°<β(β’)<90°. And the larger the β or β’ is, the larger the mode I component at crack tip is.
Mixed mode loading angles β are determined by loading holes in the arc loading champs, just shown in Fig. 4.
6.Figure 1: why the curved specimens were used instead of a flat specimens to assess fracture behavior?. Details of the curvature are not presented in the figure.
Response: We are so sorry that the description in the original manuscript is unclear, the descriptions about the specimen and loading device have been improved (Fig. 3). The fracture specimen shown in Fig. 3(a) is a flat specimen with a single-edged crack; and the fracture specimen is connected to the curved clamp, which is aim to obtain the mixed mode loading. The whole loading device is shown in Figs. 3(b) and 3(c). The test loading device is shown in Fig. 4.
7.Figure 14 could be explained in more detail.
Response: More descriptions about Fig. 16 (Fig. 14 in the original manuscript) have been added in Section 4.3.
8.The discussion and conclusion sections are clearly presented. However, the scope for future work may include at the end of the introduction.
Response: The scope for future work is include in the conclusion. Thanks for the suggestions.

Reviewer 2 Report
The manuscript " Study on I-II-III mixed mode fracture behavior and II-III coupling effect” has as its main objective to analyse the fracture criterion and II-III coupling effect on PMMA samples. as the research object, focusing on the study of I-II mixed mode crack, I-III mixed mode crack, II-III mixed mode crack and I-II-III mixed mode crack. An experimental-FE approach was employed to extract the properties of material under I-II and I-III mixed mode loading. The findings were compared and conclusions were made.
The manuscript is original and has merit, but it needs to be improved.
The abstract does not clearly state the material used in this research, methodology, results, and conclusions.
Page 2 The paragraph is too long. Please split the text into two or more paragraphs.
Correct the formatting mistakes on units and references citations.
Compare the results with those found in literature.
The figure’s size and resolution must be improved.
All items to be corrected were highlighted in yellow in the pdf document.

Author Response
Dear editor:
Thank you for giving another chance to improve the paper. The responses to reviewers are listed as below.
Reviewer #2
The manuscript " Study on I-II-III mixed mode fracture behavior and II-III coupling effect” has as its main objective to analyse the fracture criterion and II-III coupling effect on PMMA samples. as the research object, focusing on the study of I-II mixed mode crack, I-III mixed mode crack, II-III mixed mode crack and I-II-III mixed mode crack. An experimental-FE approach was employed to extract the properties of material under I-II and I-III mixed mode loading. The findings were compared and conclusions were made.
The manuscript is original and has merit, but it needs to be improved.
- The abstract does not clearly state the material used in this research, methodology, results, and conclusions.
Response: The abstract has been improved according to the suggestion, with the material, methodology, results and conclusions included.
- Page 2 The paragraph is too long. Please split the text into two or more paragraphs.
Response: The text in Page 2 has been split into two paragraphs, and the descriptions have been improved.
- Correct the formatting mistakes on units and references citations.
Response: Thanks for the suggestions. We have tried our best to correct the formatting mistakes of units throughout the paper. And all the references citations have been checked.
- Compare the results with those found in literature.
Response: The results obtained in this paper have been compared with theoretical criterion, which has been verified by tests. the relative researches have been cited in Figs. 10 and 17.
- The figure’s size and resolution must be improved.
Response: We have tried our best to improve all the figures including size and resolution in the paper.
- All items to be corrected were highlighted in yellow in the pdf document.
Response: Thanks for the suggestion again, all the items highlighted in the document have been corrected.

Reviewer 3 Report
Study on I-II-III mixed mode fracture behavior and II-III coupling effect – M. Xinting et al.
General Comments: The current work attempts to describe the coupling between the three modes under the ambit of LEFM. A representative specimen using PMMA employing titled CTS configurations have been presented. While the present paper describes the coupling phenomenon in elastic materials, it fails to acknowledge the same problem relevant in laminated structures. My specific comments are provided below.
Specific Comments:
1. Improve the introduction section to provide the mixed-mode fracture behavior. For an uninformed reader, the equations do not make sense.
2. Introduce the variables in Eq. (1) right below them.
3. Also introduce the pertinence of mixed-mode problems in homogenous, elastic medium and how they differ from layered elastic media doi: 10.1177/0021998317749714, doi: 10.1177/1099636218777964, doi: 10.2514/1.J056039, and doi: 10.1177/1099636218788223 where the mode mixity is explicitly defined using the phase angle, that dictates crack kinking, propagation path, initiation etc. This same definition can be easily applied on homogenous elastic medium, such as the material considered in this study.
4. Line 95: “consider [] the …”
5. How is it calculated that a tilt angle beta = 0 deg gives mode II condition? Without the definition of a mode mixity phase angle, obtained either using closed form expressions or numerical approaches the mixed mode component cannot be calculated. See doi: 10.1177/1099636218777964, and doi: 10.1016/j.engfracmech.2018.06.036.
6. It is extremely important to capture the tilt angle of the specimen and its corresponding mode-mixity, as this is sensitive in decoupling the mode I-II mixed mode fracture behavior.
7. This is a suggestion: perhaps, the FE-model could be simplified using MPE elements and rigid body connectors instead of modeling the entire loading block.
8. Was the Crack opening displacement used from the FE-model to estimate the stress-intensity factors?
9. Elaborate if a mesh sensitivity study was performed?
10. Elaborate whether the total strain energy was computed from the FE-model and compared against experimental results?
11. Section 3 may just be titled “Mixed mode I-II and I-III Crack Propagation Behavior”. The entire paper deals with research on this particular mixed-mode behavior.
12. Was the global load-displacement behavior compared against FE-model results?
13. What is the implication of the proposed new criterion on material and geometrical properties? Is there a limit on E and h?
14. Include a recommendation section on how to use the proposed criterion and whether there are any calibration procedure required for modified specimen geometry (and material).
Moderate editing of English language style and grammar recommended.
Author Response
Dear editor:
Thank you for giving another chance to improve the paper. The responses to reviewers are listed as below.
Reviewer #3
General Comments: The current work attempts to describe the coupling between the three modes under the ambit of LEFM. A representative specimen using PMMA employing titled CTS configurations have been presented. While the present paper describes the coupling phenomenon in elastic materials, it fails to acknowledge the same problem relevant in laminated structures. My specific comments are provided below.
Specific Comments:
1. Improve the introduction section to provide the mixed-mode fracture behavior. For an uninformed reader, the equations do not make sense.
Response: The introduction of mixed mode crack has been improved (adding Fig.1).
The equations have been explained by the figure of crack tip field (Fig. 2 in the revised paper).
2. Introduce the variables in Eq. (1) right below them.
Response: The variables in Eq. (1) has been explained.
3. Also introduce the pertinence of mixed-mode problems in homogenous, elastic medium and how they differ from layered elastic media
doi: 10.1177/0021998317749714,
doi: 10.1177/1099636218777964,
doi: 10.2514/1.J056039, and
doi: 10.1177/1099636218788223 where the mode mixity is explicitly defined using the phase angle, that dictates crack kinking, propagation path, initiation etc. This same definition can be easily applied on homogenous elastic medium, such as the material considered in this study.
Response: Thanks for the recommendation. Mixed-mode problems in layered elastic media mentioned above all have been introduced in the revised paper.
The comparisons of mixed mode fracture between homogenous, elastic medium and layered elastic media will be our next study topic. Thanks for the suggestion.
4. Line 95: “consider [] the …”
Response: The sentence has been improved. Thanks for the suggestion.
5. How is it calculated that a tilt angle beta = 0 deg gives mode II condition? Without the definition of a mode mixity phase angle, obtained either using closed form expressions or numerical approaches the mixed mode component cannot be calculated. See doi: 10.1177/1099636218777964, and doi: 10.1016/j.engfracmech.2018.06.036.
Response: There were some errors and confusions in the description of mode mixity in the original paper, which have been corrected and improved in the revised paper.
It is pure mode I loading when β or β’=90°, pure mode II loading when β=0°, pure mode III loading when β’=0°, and mixed mode loading when 0°<β or β’<90°. And the larger the β or β’ is, the larger the mode I component at crack tip is. The arc loading device and loading angle β or β’ are both designed and defined based on the definition of mode II, mode III crack and mixed mode crack.
The mode mixty in this paper is defined by the ratio of mode I, mode II and mode III component at crack tip, which are obtained by FEM, and written as KIIn=KII/KI+KII+KIII), KIIIn=KIII/(KI+KII+KIII)).
The discussion of mode mixty (KIIn and KIIIn) with crack propagation and fracture toughness are shown in Fig. 8, Fig. 10 and Fig. 17.
6. It is extremely important to capture the tilt angle of the specimen and its corresponding mode-mixity, as this is sensitive in decoupling the mode I-II mixed mode fracture behavior.
Response: We agree with the reviewer’s suggestion, mode-mixity in this paper is defined as KIIn=KII/KI+KII+KIII) and KIIIn=KIII/(KI+KII+KIII)), which has an important effect on fracture behaviors. The discussion of mode mixity (KIIn and KIIIn) with crack propagation and fracture toughness are shown in Fig. 8, Fig. 10 and Fig. 17.
7. This is a suggestion: perhaps, the FE-model could be simplified using MPE elements and rigid body connectors instead of modeling the entire loading block.
Response: We agree with the reviewer’s suggestion, the FE-model can be simplified using MPE elements and rigid body connectors, which will be applied in our next researches. Thanks for the suggestion.
The loading clamp in this paper was also set as a rigid body, and contacts were set between loading clamps and specimens, which can transfer loading. And in order to help readers understand the loading situation in this article, the entire loading block was present.
8. Was the Crack opening displacement used from the FE-model to estimate the stress-intensity factors?
Response: The stress-intensity factors was estimated from the maximum loading by test, according to the method of determining the plane strain fracture toughness KIC in GB/T 21143-2014. The description has been added in the revised manuscript.
9. Elaborate if a mesh sensitivity study was performed?
Response: The mesh sensitivity study has been added. Thanks for the suggestion.
10. Elaborate whether the total strain energy was computed from the FE-model and compared against experimental results?
Response: In the linear elastic fracture mechanics, energy release rate G is equal to stress intensity factor K in mechanics, there is a definite relationship between G and K. In this paper, K-factor has been obtained by test, and which has a good agreement with theoretical results in linear elastic fracture mechanics (shown in Fig. 10). So, we don’t compute the strain energy ahead of crack tip as the fracture parameter.
While strain energy is an important factor during fatigue crack growth, which will be studied in the future work we are devoting to. Thank you so much.
11. Section 3 may just be titled “Mixed mode I-II and I-III Crack Propagation Behavior”. The entire paper deals with research on this particular mixed-mode behavior.
Response: The title of Section 3 has been corrected according to the reviewer’s suggestion.
12. Was the global load-displacement behavior compared against FE-model results?
Response: The comparisons of load-displacement behaviors between FEM and tests have been present in Fig. 8. The discussions and explanations have been added in detail.
13. What is the implication of the proposed new criterion on material and geometrical properties? Is there a limit on E and h?
Response: The new criterion can be used to predict crack growth angle of mixed mode crack, which considers the II-III coupling effect at crack tip. The criterion is not limit to the specimen and material studied in this paper, which applies to any brittle mixed mode fracture, and related to mode mixity ahead of crack tip (KIIn and KIIIn).
Are the factors of E and h related to crack tip field or any brittle material? If yes, the new criterion has no limit to use.
14. Include a recommendation section on how to use the proposed criterion and whether there are any calibration procedure required for modified specimen geometry (and material).
Response: The proposed criterion is related to mode mixity factor, like KII/(KI+KII+KIII) and KIII/(KI+KII+KIII). For specimen with any geometry and material, if stress intensity factors (KI, KII, KIII) at crack tips are determined, crack growth paths will be fixed.
As the reviewer’s suggestion, more descriptions about the criterion have been added.

Round 2
Reviewer 3 Report
The manuscript may be accepted in its revised format. Make sure description Figure 4 is corrected during editing.